# The Mediating Role of Positive Orientation in the Relationship between Loneliness and Meaning in Life

**DOI:** 10.3390/ijerph19169948

**Published:** 2022-08-12

**Authors:** Dominik Borawski

**Affiliations:** Department of Psychology, Jan Kochanowski University, 25-029 Kielce, Poland; dborawski@ujk.edu.pl

**Keywords:** loneliness, meaning in life, positive orientation

## Abstract

(1) Background: Previous research revealed that increased loneliness resulted in decreased meaning in life (MIL). Little is known, however, about the underlying mechanisms of this relationship. The aim of the study was to determine if the set of positive evaluations of oneself, one’s life, and one’s future discussed by Caprara as positive orientation (POS) was a mediator between loneliness and MIL. (2) Methods: A sample of 304 Polish participants aged 19–45 (*M* = 25.61 years, *SD* = 6.1) completed the De Jong Gierveld Loneliness Scale, the Presence subscale of the Meaning in Life Questionnaire, and the Positivity Scale. (3) Results: The study showed that loneliness was negatively associated with POS and MIL, while POS and MIL were positively correlated with each other. It also turned out that POS partially mediated the relationship between loneliness and MIL. (4) Conclusions: The results suggest that, by inhibiting POS, loneliness makes it difficult to perceive life as meaningful. The interrelationships between loneliness, POS, and MIL are discussed in light of the evolutionary theory of loneliness.

## 1. Introduction

The links between interpersonal relations and meaning in life (MIL) have been researched for nearly two decades. The results of previous studies show that both chronic loneliness and situational social rejection lead to decreased MIL [1]. Little is known, however, about the underlying mechanisms of this relationship. As an attempt to fill this gap, the present study was inspired by research and theories suggesting that possible mediators between perceived quality of social connections and MIL can be sought among the cognitive evaluations of the self and the world [1,2]. Following this lead, the author set out to determine if positive beliefs about oneself, one’s life, and one’s future (i.e., positive orientation, POS) were a mediator between loneliness and MIL. 

### 1.1. Loneliness and Meaning in Life

People consider life meaningful if they can see purpose (i.e., aims and aspirations), coherence (i.e., the comprehensibility of life and sense to be made of it), and significance (i.e., value beyond the trivial or momentary) in different domains of their existence [3]. What most people point to as the most important source of meaning is close relationships with others [4,5]. Social connections allow an individual to feel part of a larger symbolic community and to expand the limits of their own self; in a way, they ensure symbolic immortality [6]. Above all, however, they make it possible to perceive the surroundings as predictable and safe, which in turn is conducive to expanding the zone of proximal development, formulating plans, and engaging in purposeful actions. This is shown by studies suggesting, rooted in one’s history of social relationships, that secure attachment provides a foundation for the sense that life is meaningful [7]. By contrast, the non-satisfaction of the need for belongingness impedes the perception of life in terms of purpose and coherence. This is well illustrated by the results of research on attachment and loneliness. Both anxiety and avoidant attachment has been found to correlate negatively with MIL [8]. Similarly, loneliness, defined as subjective dissatisfaction with the quality of one’s interpersonal relations, was negatively associated with the meaning domain of well-being [9]. Furthermore, loneliness proved to be a better predictor of MIL than depression, pessimism, and negative mood [1]. The cause-and-effect nature of the relationship between perceived quality of social relationships and MIL is evidenced by the results of the experimental studies conducted to date. Twenge et al. [10] found that the experience of social rejection led to a deconstructed state of mind, whose indicators included a decrease in meaningful thought. Other research showed that being rejected in an interactive computer game called Cyberball led to the contextual deprivation of basic needs, including the need for a meaningful existence, and to a decrease in the general evaluation of life as meaningful [1].

### 1.2. Mediating Role of Cognitive Evaluations of Oneself, One’s Life, and One’s Future

There are both theoretical and empirical reasons to seek the key mechanisms of the relationship between loneliness and MIL among variables reflecting the cognitive evaluations of the self and the world. For example, meaning-making models suggest that self-beliefs constitute cognitive frameworks making it possible to interpret life events and to perceive them as more or less meaningful [11]. In the case of difficult and surprising experiences, these beliefs are shaken, which leads to a situational loss of meaning in life (lasting until the reconstruction of mental representations). Accordingly, in the context of social relationships, attachment theories posit that internal mental models of the self, others, and the world are mediators of the relationship between early interpersonal experience and MIL [2]. It seems that a similar mechanism can be found in the case of loneliness. According to the evolutionary theory of loneliness, subjective social isolation activates the primal anxiety-filled self-preservation mode of cognition, marked not only by a negativistic pattern of evaluating others but also by a pessimistic evaluation of oneself and one’s past, which may affect the sense of individual MIL [11,12]. Moreover, existing studies suggest that a particularly significant factor mediating the quality of interpersonal relations and MIL is the way a person perceives themselves. For example, self-compassion mediated the relationship between attachment and purpose in life [13], self-authenticity was an important underlying mechanism between loneliness and meaning [9], and Stillman found that the key mediator between loneliness and decreased MIL was self-worth operationalized as global self-esteem [1]. Following this lead in the current study, the author set out to determine if POS as a variable that “mostly concerns how people construe themselves within the world” [14] (p. 56) was a mediator in the relationship between loneliness and MIL.

### 1.3. Positive Orientation as a Potential Underlying Mechanism

As a response to Beck’s work on the cognitive triad of beliefs that accompany depression (i.e., negative evaluations concerning the self, the world, and the future), Caprara [15] proposed the opposite triad, constituting the cognitive core of happiness. The beliefs he presented as POS include evaluations of oneself (i.e., self-esteem), one’s future (i.e., optimism), and one’s life (i.e., life satisfaction); they were originally referred to as positive thinking and cognitive orientation. These three types of beliefs are the foundation of social cognitive processing and are responsible for the tendency to perceive life experiences in a positive way. It is worth noting that, unlike Beck’s triad, POS does not include beliefs about the world, and the cognitive evaluations it concerns are focused on the self. Life satisfaction relates to the affective balance of one’s own life and comprises an evaluation of one’s past and present, optimism is the expectation that one’s desirable life scenarios will come true in the future, and self-esteem refers to the general evaluation of self-worth. Moreover, Oleś et al. [14] found that POS was strictly associated with another self-belief—namely, self-efficacy. POS can therefore be viewed, above all, as the common basis for positive judgments about oneself and as reflecting the way people define themselves within the world [14]. 

Although POS is a primal and relatively stable disposition, in his theory Caprara assumes the possibility of modifying its level. This possibility was confirmed by the results of recent studies, which showed that both individual factors associated with personal success [16,17] and the way of perceiving the social environment influenced the level of POS [18]. In the present study, it was hypothesized that POS would be inhibited by loneliness, which was a primary affective experience stemming from the deprivation of the basic need for affiliation, and that this would translate into lower MIL. The hypothesis concerning the mediating role of POS in the relationship between loneliness and meaning in life (see Figure 1) is justified by studies showing that loneliness and a sense of belongingness are predictors of POS [19,20] and by research results indicating that both POS and its components are associated with MIL [21]. Even stronger arguments in favor of this line of reasoning are provided by experimental and longitudinal studies. Cacioppo et al. [22] found that experimentally induced loneliness lowered the levels of self-esteem and optimism, while Shankar et al. [23] reported that loneliness longitudinally predicted lower subjective well-being (a variable very close to life satisfaction). The results of other experimental studies suggest that specific POS components operationalized as situational manipulations contribute to increased MIL. These components are positive self-evaluations [24], positive concepts associated with positive emotions and life satisfaction [25], and anticipating happiness [26]. Based on the above, it is reasonable to suspect that the cognitive evaluations making up POS mediate the relationship between loneliness and meaning—namely, that loneliness undermines self-worth, positive evaluation of one’s life, and optimistic view on one’s future, thus impeding the perception of meaning in one’s existence.

## 2. Materials and Methods

### 2.1. Participants and Procedure

To estimate the required sample size, the researcher conducted a Monte Carlo power analysis for mediation models using Schoemann, Boulton, and Short’s [27] algorithm. Based on the results of previous studies, moderate correlations were expected between loneliness and both POS [19] and MIL [1], and a high correlation was predicted between POS and MIL [18]. The analysis was performed with 1000 replications, 2000 Monte Carlo draws per replication, and a 95% confidence interval. The results suggested that, with an alpha of 0.05 and a power of 0.95, a sample of 81 participants was required to detect the effects of the predicted size. 

Eventually, the author collected data from 308 participants. However, the statistical analysis included data from 304 participants (4 cases were rejected during data screening; see details in the Statistical Analyses section), who were Poles aged 19 to 45 (*M* = 25.61 years, *SD* = 6.1). The majority of the sample (65.5%) were women; 62.5% of the participants had secondary education, 20.1% had an MA or higher degree, 15.1% had a BA degree, and 2.3% had vocational education. In regards to place of residence, 56% of the participants lived in the countryside or in small towns (with up to 20,000 residents), while 44% lived in medium-sized (between 20,000 and 100,000 residents) or large cities (between 100,000 and 500,000 residents). The study was conducted in Poland, in the Świętokrzyskie and Podkarpackie provinces, as a paper-and-pencil survey. Convenience sampling was applied. Data were collected from March 2019 to December 2019 by appropriately trained students, who administered the measures to adult respondents recruited through social network sites and word-of-mouth advertising. All participants were briefed about the purpose of the study and gave their informed consent to participate in it. Confidentiality and anonymity were ensured. Participants did not receive any remuneration for taking part in the study. They completed the questionnaires individually or in small groups.

### 2.2. Measures

#### 2.2.1. Loneliness

The De Jong Gierveld Loneliness Scale [28], adapted into Polish by Grygiel et al. [29], was used to assess loneliness, understood as a general sense of dissatisfaction with the quality of one’s social relationships. The measure consists of 11 items (e.g., “I miss having really close friends”). Participants responded using a 5-point Likert scale (1 = *strongly agree* to 5 = *strongly disagree*). 

#### 2.2.2. MIL

The Presence subscale of the Meaning in Life Questionnaire [30], adapted into Polish by Kossakowska et al. [31], was used to assess the degree to which the participants perceived their current lives as meaningful. The measure consists of 5 items (e.g., “My life has a clear sense of purpose”) rated on a 7-point Likert scale (1 = *absolutely untrue* to 7 = *absolutely true*).

#### 2.2.3. POS

POS was measured with the Positivity Scale [32], adapted into Polish by Laguna et al. [33]. The instrument has a one-factor structure and consists of 8 items concerning self-esteem (e.g., “I feel I have my things to be proud of”), life satisfaction (e.g., “I am satisfied with my life”), and optimism (e.g., “I look forward to the future with hope and enthusiasm”). Participants responded using a 5-point Likert scale (1 = *strongly disagree* to 5 = *strongly agree*). 

### 2.3. Statistical Analyses

Before the beginning of the main analysis, data were screened for potential errors in the expected range of values and for any indicators of careless answers. Data from four respondents were excluded when it turned out that they had ignored even up to 50% of the items. After the exclusion of these cases, missing data accounted for 3.16% of all answers given by participants. The missing values were missing completely at random and were therefore imputed using the EM (expectation–maximization) algorithm [34]. Data were also screened for multivariate outliers. However, as no data point exceeded a Cook’s distance of 1, all remaining observations (*N* = 304) were included in the main statistical analysis.

Data analyses were performed in several steps. Descriptive statistics and correlations between the variables were computed using IBM SPSS Statistics software (version 26, PS IMAGO PRO 6.0, Predictive Solutions, Krakow, Poland). Mediation was tested using the PROCESS macro [35] with the bootstrapping method, which relied on bias-corrected confidence estimates (10,000 bootstrapped resamples). In the mediation analysis, participants’ sex (0 = male, 1 = female) and age were included as covariates. Additionally, the researcher applied AMOS software to calculate fit indices in order to compare potentially competing models with different sequences of variables.

## 3. Results

### 3.1. Preliminary Analyses

Descriptive statistics, reliability coefficients, and correlations between the variables are presented in Table 1. Means and standard deviations for all the variables were similar to those already reported in other studies conducted in Poland [36,37]. The normal distribution of the variables was supported, as the values of skewness and kurtosis did not exceed 1 [38]. No evidence of multicollinearity was found (TOL > 0.02, VIF < 4). 

Zero-order correlations indicated that loneliness was negatively associated with POS and MIL, while POS and MIL were positively correlated with each other. 

### 3.2. Testing for Mediation Effect 

As shown in Table 2, loneliness was associated with POS (b = −0.34, SE = 0.04, *p* < 0.001), while POS was associated with MIL (b = 0.84, SE = 0.10, *p* < 0.001). The indirect effect of loneliness on MIL through POS was found to be significant, as the 95% confidence interval did not include zero (b = −0.28, SE = 0.05, 95% CI = [−0.39, −0.19]), which supports H1. Moreover, the direct effect of loneliness on MIL remained significant (b = −0.30, SE = 0.08, *p* < 0.001), which indicated that POS partially mediated the relationship between loneliness and MIL. The indirect effect accounted for 48.94% of the total effect.

However, because both loneliness and POS can be viewed as basic dispositions and because, theoretically, it was also possible that POS was an antecedent of interpersonal functioning [39], an additional alternative model was tested, with loneliness as a mediator between POS and MIL. To compare these two competing models, the author used the Akaike Information Criterion (AIC), with smaller values indicating better model fit, and the Expected Cross-Validation Index (ECVI), with smaller values indicating a greater potential for replication. It turned out that for the original model, with POS as a mediator, the values of AIC and ECVI were 20.97 and 0.07, respectively, while for the alternative model, with loneliness as a mediator, the respective values were 73.21 and 0.24. These values of AIC and ECVI suggest that the original model is better fitted to the data and has greater potential for replication than the alternative one.

## 4. Discussion

In the current study, it was hypothesized that POS, the common basis for positive judgments about oneself, one’s life, and one’s future, would function as a mediator between loneliness and MIL. 

A moderate negative correlation was found between loneliness and the perceived presence of MIL, which is in line with previous studies on these variables [1,9]. Furthermore, it turned out that POS partially mediated the relationship between loneliness and MIL. This means that, by inhibiting POS, loneliness makes it difficult to perceive life as meaningful. This result corresponds both with studies on the negative cognitive evaluations that accompany loneliness [40] and with the research that shows the association of POS with MIL [21]. An argument that supports the interpretation of loneliness as an antecedent of POS is the results of earlier studies, suggesting a cause-and-effect relationship between loneliness and the variables that make up the POS construct. Cacioppo et al. [22] found that experimentally induced loneliness resulted in decreased beliefs such as self-esteem and optimism. Moreover, if POS is considered the opposite of the depressive cognitive triad, these results are consistent with both cross-sectional and longitudinal studies showing that loneliness predicts depressive symptoms [41,42]. The second key element constituting the mediation effect that the author has found—namely, the relationship between POS and MIL—is consistent with the research results suggesting that POS is the cognitive foundation of MIL [21]. 

The interrelationships between loneliness, POS, and MIL can be interpreted in light of the evolutionary theory of loneliness [43]. According to this theory, loneliness is associated with a strong sense of insecurity, which leads to the activation of an anachronistic survival mechanism characterized by heightened sensitivity to social threats and negative evaluations of oneself. The consequences of this mechanism include a decrease in self-esteem, predicting one’s own future in terms of threat, and an anxiety-filled perception of other people’s intentions [22]. Thus, by enhancing motivation for short-term self-preservation, loneliness inhibits the meaning-making features of POS. Being vigilant to social threats makes it difficult to focus on long-term goals, which are highly important in shaping the meaning of one’s life. Furthermore, given that one of the main components of POS is self-esteem, the results of the present study also correspond with those reported by Stillman et al. [1], who find that the need for self-worth is the key explanatory mechanism between loneliness and MIL.

The present findings should be interpreted in light of certain limitations. First, the convenience sampling applied by the author, resulting in a sample of predominantly young and female participants, limits the generalizability of the results. The relationships between loneliness, POS, and meaning discussed above may be different for men, older adults, and people from countries more ethnically diverse than Poland. For example, Krause and Rainville [44] found that sense of meaning was particularly important for individuals in late life and that older adults derive meaning from social relationships to a greater extent than younger ones. Second, even though the proposed mediation model was better fitted to the data than the alternative one, causality cannot be inferred because the present study was cross-sectional. Therefore, the presented relationships between loneliness, POS, and MIL require further detailed research, particularly longitudinal and experimental studies. Finally, the current investigation could be replicated with other variables controlled for that may also mediate the relationship between loneliness and MIL. Recent studies suggest that such variables could be authenticity [9] and mindfulness [37]. 

## 5. Conclusions

Loneliness is a well-documented predictor of decreased MIL. However, studies are still lacking that would identify the underlying mechanisms of this relationship. The current study, constituting an attempt to fill this gap, was inspired by research and theories suggesting that a possible explanation of this relationship should be sought in the cognitive aspects of perceiving the world, particularly in self-beliefs. The results of the current investigation demonstrated that POS, understood as positive judgments concerning oneself, one’s life, and one’s future, was a mediator in the relationship between loneliness and MIL. This finding suggests that loneliness undermines self-worth, positive evaluation of one’s life, and an optimistic view of one’s future, thus impeding the perception of meaning in one’s existence. The study makes a useful contribution to the growing literature on the loneliness–meaning link, shedding new light on the cognitive mechanism behind the loss of meaning as a consequence of subjectively perceived social isolation.

## Figures and Tables

**Figure 1 ijerph-19-09948-f001:**
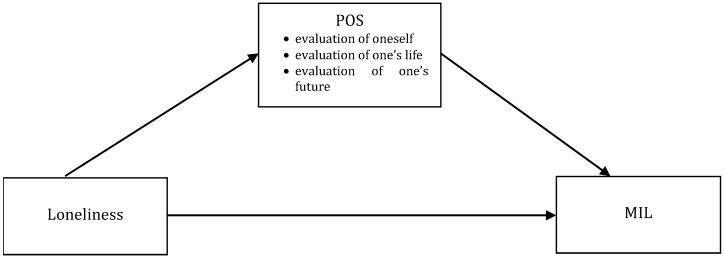
Proposed simple mediation model.

**Table 1 ijerph-19-09948-t001:** Descriptive statistics, Cronbach’s alphas, and zero-order correlations between the variables.

Variable	*M*	*SD*	α	1	2	3	Skewness	Kurtosis
1. Loneliness	2.31	0.71	0.86	–			0.16	−0.51
2. POS	3.64	0.55	0.76	−0.42 ***	–		−0.35	−0.12
3. MIL	4.65	1.08	0.83	−0.37 ***	0.52 ***	–	−0.26	−0.12

Note. POS = positive orientation; MIL = meaning in life; *** *p* < 0.001.

**Table 2 ijerph-19-09948-t002:** Simple mediation model.

Predictor	POS			MIL		
	*b*	*SE*	*t*	*b*	*SE*	*t*
Constant	4.27	0.18	23.43	1.66	0.55	3.02 *
Sex	−0.03	0.06	−0.50	−0.02	0.11	−0.17
Age	0.01	0.01	1.64	0.02	0.01	2.90 **
Loneliness	−0.34	0.04	−8.19 ***	−0.30	0.08	−3.63 ***
POS	−	−	−	0.84	0.10	8.14 ***
	*R*^2^ = 0.187 *F* (3, 300) = 22.94 ***			*R*^2^ = 0.316 ^p*F* (4, 299) = 34.61 ***		

Note. POS = positive orientation; MIL = meaning in life; * *p* < 0.05. ** *p* < 0.01. *** *p* < 0.001.

## Data Availability

The anonymized data are available from the author upon reasonable request.

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
