# Peer review of "The Mediating Role of Positive Orientation in the Relationship between Loneliness and Meaning in Life"

_ijerph, 2022, doi:10.3390/ijerph19169948_

Round 1
Reviewer 1 Report
This is a well-written study on the mediating factors between loneliness and meaning in life among midlife adults in Poland.
1. Suggest first sentence remove “for at least a dozen years or so” in favor of “for nearly two decades.”
2. Remove personal language like “I decided” and “I hypothesized” and replace with “the researcher determined” or “hypothesized” throughout manuscript.
3. Suggest revising the proposed mediation model (figure 1) on page 3 which is confusing because it has been too simplified. If it is hypothesized that POS is mediating the relationship between loneliness and meaning in life, then there should be no direct line drawn between loneliness and MIL at the bottom. A mediating effect would instead be centered between loneliness and MIL like this: LONE à POS à MIL. It might also be a more useful model if you wrote the the POS components inside the POS box at the center of the graphic (evaluation of oneself, evaluation of one’s future, evaluation of one’s life). Additionally, your discussion on p. 6 details more specific pathways of interest that you may consider adding to your model, such as loneliness leading to anxiety and insecurity, which decreases self-esteem, leading to lowered POS and MIL. And maybe this figure should be moved to the “discussion” section?
4. Materials and methods on pages 3-4 need more description. How did you and the research assistants collect these data, was it in-person or online surveys? How were respondents selected? You mention convenience sampling, but this needs to be described more fully, was it word-of-mouth from other participants? Were the trained assistants students, paid local employees, some other type? Who granted permission for the study and approved the informed consent documents, a university or hospital? During what months and year was data collected? Did this sample include Poles across the country, or just in a certain area? Which regions of the country were sampled and how? Are all Poles considered ethnically the same, or are there ethnic/racial divisions to report?
5. Please consider adding a table to page 5 that gives descriptive characteristics of the participants, so the reader can see the breakdown of participants by age, gender, ethnicity/race, educational attainment, place of residence, and any other measured features of your sample.
6. The paragraph at the bottom of page 6 on the study limitations should include a sentence or two about how a sample of mostly younger Polish women limits generalizability of the study. Please make mention of the research that demonstrates how loneliness, POS, and meaning in life may be different for men, older adults, and/or people of different ethnic backgrounds.
7. The introduction and conclusion are great!
Author Response
Please see the attached response letter. Thank you very much.

Reviewer 2 Report
This paper is a contribution to the analysis of the role of Positive Orientation as mediator of the relationship between Loneliness and Meaning in Life, contributing to clarify the mechanisms of this relationship.
Method, instruments, sample and statistical analysis is adequate. Results and it's discussion are relevant.
However, the article could be improved in some aspects:
1. It would be important to indicate the internal consistency of the measures used in this investigation.
2. Some limitations are mentioned in the discussion, but it is not mentioned that it will be important to study other variables that may mediate the relationship between Loneliness and Meaning in Life.
3. It seems to us that the article should include references to previous studies published in “Int. J. Environ. Res. Public Health”.
4.English should be revised and “I” should be replaced by “were”, for example in “I conducted” (p.3) or “I expected” or “I collected” (p.4).
Author Response

(The authors gave the same response as above.)

Reviewer 3 Report
1. Am not sure why sections in the abstract are numbered.
Additional comments:
- It is timely, interesting, and well written
- It addresses an issue [loneliness and MIL] which are intuitively related, and affect each other
- It, interestingly, connects POS to loneliness as well as to MIL, which is significantly related to it.
- While the Discussion is not very long, I found it interesting and addressing teh issues that the manuscript dealt with.
Author Response

(The authors gave the same response as above.)
